# A Biocontrol Strain of *Serratia plymuthica* MM Promotes Growth and Controls Fusarium Wilt in Watermelon

Zhaoyu Li [1,†], Jinxiu Ma [1,†], Jiajia Li [1], Yinglong Chen [2], Zhihong Xie [3,*], Yongqiang Tian [1,*], Xu Su [4], Tian Tian [1] and Tong Shen [1]

1   School of Biological and Pharmaceutical Engineering, Lanzhou Jiaotong University, Lanzhou 730070, China; lizy@mail.lzjtu.cn (Z.L.); xmm35378@gmail.com (J.M.); lijiajia@mail.lzjtu.cn (J.L.); tiantian@mail.lzjtu.cn (T.T.); shentong@mail.lzjtu.cn (T.S.)
2   The UWA Institute of Agriculture, School of Agriculture and Environment, The University of Western Australia, Perth, WA 6009, Australia; yinglong.chen@uwa.edu.au
3   National Engineering Research Center for Efficient Utilization of Soil and Fertilizer Resources, College of Resources and Environment, Shandong Agricultural University, Tai'an 271018, China
4   Laboratory of Biodiversity Formation Mechanism and Comprehensive Utilization of the Qinghai-Tibet Plateau in Qinghai Province, Qinghai Normal University, Xining 810008, China; 2025086@qhnu.edu.cn
*   Correspondence: zhihongxie211@163.com (Z.X.); tianyq@mail.lzjtu.cn (Y.T.)
†   These authors contributed equally to this work.

**Abstract:** Fusarium wilt, caused by *Fusarium oxysporum* f. sp. *niveum* (FON), is a predominant and devastating soil-borne disease that results in significant yield losses in watermelon cultivation. In this study, a strain MM isolated from the herbage rhizosphere soil, exhibited an inhibition rate of 65.46% against FON, leading to mycelial collapse, atrophy, and deformation. In pot experiments, strain MM effectively controlled Fusarium wilt of watermelon, showing a control efficacy of 74.07%. Through morphological observation and 16S rDNA gene sequencing, strain MM was identified as *Serratia plymuthica*. Additionally, *S. plymuthica* MM demonstrated antagonistic activity against eight plant pathogens, indicating that MM had broad-spectrum antifungal activity. The strain also exhibited the ability to synthesize siderophores and indole acetic acid (IAA), both of which are growth-promoting compounds. Moreover, strain MM secreted various extracellular enzymes, including protease, chitinase, β-glucanase, and cellulase. This ability allowed *S. plymuthica* MM to readily colonize watermelon roots and promote seedling growth. Inoculation with *S. plymuthica* MM increased the activity of PAL, POD, PPO, and CAT enzymes associated with watermelon defense. Furthermore, qRT-PCR analysis revealed up-regulation of *LOX*, *POD*, *PAL*, *ClPR3*, and *C4H* genes, which are related to plant disease resistance. The results indicated that *S. plymuthica* MM enhances watermelon plants' resistance to FON by activating the JA, SA, and shikimic acid phenylpropanoid–lignin synthesis pathways. Gas chromatography–mass spectrometry (GC-MS) analysis of *S. plymuthica* MM culture supernatant identified piperazinedione, pyrrolo[1,2-a]pyrazine-1,4-dione, and octadecenamide as the main antimicrobial substances. Overall, *S. plymuthica* MM shows promise as a biocontrol agent against Fusarium wilt of watermelon, suggesting its potential for the development of a new biocontrol agent.

**Keywords:** Fusarium wilt of watermelon; *Serratia plymuthica*; biological control; FON; growth promotion

## 1. Introduction

Fusarium wilt of watermelon, caused by *Fusarium oxysporum* f. sp. *niveum* (FON), is a destructive soil-borne disease that poses a significant threat to watermelon cultivation worldwide [1].Under severe infection conditions, Fusarium wilt can cause up to 100% yield losses in watermelon production [1]. FON infects watermelon roots through root hairs or root wounds and spreads within vascular bundles, disrupting the transport tissues and blocking water flow, eventually leading to plant wilting and death [2]. Due to long survival period of its chlamydospore in soil, FON is highly persistent and difficult to control, posing a major hindrance to watermelon production [3].

Current methods for managing Fusarium wilt of watermelon include host resistance, grafting, soil fumigation, and chemical applications [4]. However, breeding for multilocus disease-resistance is time-consuming, and grafting may affect watermelon quality. Soil fumigation can eliminate soil pathogens but also harm the ecological balance of soil microorganisms. Chemical fungicides are commonly used for FON control, but they come with drawbacks such as environmental pollution and pesticide resistance [5]. Therefore, there is a need to explore effective and environmentally friendly approaches for managing Fusarium wilt of watermelon, such as biological control.

In recent years, the use of microorganisms as biocontrol agents for plant diseases has gained attention and become a research focus [6,7]. Numerous studies have demonstrated that antagonistic microorganisms can suppress the growth of plant fungal pathogens and promote plant growth [8–11]. Several biocontrol agents, including *Bacillus subtilis*, *Fluorescent pseudomonads*, *Paenibacillus polymyxa*, *Pseudomonas chlororaphis*, *Bacillus amyloliquefaciens*, *Bacillus velezensis,* and *Trichoderma asperellum*, have been employed for the biological control of Fusarium wilt of watermelon [12,13]. In addition, *Serratia plymuthica* has demonstrated efficacy as biological control agent and a plant-growth-promoting bacterium in management of various plant diseases, including bean root rot, pepper phytophthora blight, black leg potato, fire blight of rosaceous plants, and potato soft rot [14–16]. It produces a range of antagonistic substances such as pyrrolnitrin, haterumalides, zeamine, andrimid, serratamid, volatiles, and enzymes to control plant diseases as well as growth-promoting substances like siderophores and IAA to enhance plant growth [17–19]. Moreover, *S. plymuthica* exhibits rapid growth and survives in diverse conditions, making it suitable for large-scale production [20].

Although there has been some research on the antagonistic bacteria controlling Fusarium wilt of watermelon, the use of *Serratia plymuthica* as a biocontrol agent against Fusarium wilt of watermelon has not been reported. In this study, antagonistic bacteria *S. plymuthica* strain MM was isolated from the rhizosphere soil of herbage, and its biological control and plant-growth-promotion activities were investigated. Furthermore, the study evaluated the antifungal and growth-promoting mechanisms of strain MM through assessing defense enzyme activities, analyzing the expression of defense-related genes, detecting active substances in the fermentation supernatant, and estimating the presence of plant-growth-promotion substances. These findings lay a theoretical foundation for the development of *S. plymuthica* MM as a biological control agent with the potential to control Fusarium wilt and promote watermelon growth.

## 2. Materials and Methods

### 2.1. Isolation and Antagonism Test

A total of 10 herbage rhizosphere soil samples were collected from the Sanjiangyuan area (38°8′ N 101°23′ E) in Qinghai Province, which is situated at an elevation of 3956 m above sea level. The soil samples were obtained from the 15 cm depth layer near the herbage. The isolation of antagonistic bacteria was performed following the method described by Zheng et al. [21]. All bacteria were isolated from individual colonies on LB plates and stored at −4 °C for subsequent tests.

The antagonistic activity against FON was evaluated using the plate confrontation culture method [22]. Sterile water was used as the control, and each treatment was replicated three times. The antagonistic activity of the isolated bacteria was determined by calculating the inhibition rate using the following formula: Inhibition rate (%) = [D1 − D2/D1 × 100], where D1 represents the diameter of FON in the control group, and D2 represents the diameter of FON in the treatment group.

### 2.2. Identification and Inhibition Spectrum of Isolate MM

The strain MM was identified using both morphological and molecular methods [22]. The strain MM was cultured on LB medium at 28 °C for 24 h, and the shape, size, and color

of the individual colony were observed. Additionally, a single colony was stained using Gram's method and examined under a microscope.

The genomic DNA of strain MM was extracted using a DNA extraction kit (Sangon, Shanghai, China) following the manufacturer's instructions. The 16S rDNA sequence of the MM was amplified by PCR using universal primers (Table S1) [22]. The PCR products were sent to Beijing Huada Gene Company for sequencing. The resulting sequence was analyzed using BLAST on the NCBI website (https://www.ncbi.nlm.nih.gov/ (accessed on 29 April 2021). The phylogenetic tree was constructed using the neighbor-joining method of MEGA 6.0 software with bootstrap analysis of 1000 replicates.

To assess the inhibition spectrum of strain MM, eight plant pathogenic fungi were used, namely isolates of *Botrytis cinerea*, *Fusarium graminearum*, *Colletotrichum gloeosporioides penz*, *Fusarium proliferatum*, *Fusarium oxysporum*, *Fusarium moniliforme*, *Fusarium solani*, and *Fusarium avenaceum*, which were provided by the Gansu Province Plant Bio-pesticide Research Center at Lanzhou Jiaotong University, Lanzhou, China. The antagonistic activity of strain MM against tested fungi was calculated using the method described in Section 2.1. Each treatment was replicated three times.

### 2.3. Effect of MM on Spore Germination and Mycelial Morphology of FON

FON was cultured on the PDA plates at 28 °C for 5 days. Mycelial plugs with a diameter of 5 mm were excised from the plate's edge and transferred to 100 mL of PDB. The broths were incubated at 28 °C and 180 rpm for 3 days. The FON cultures were then filtered through four layers of sterile gauze to obtain a conidial suspension. The conidial concentration was adjusted to $1 \times 10^5$ conidia/mL using a hemacytometer.

The strain MM was inoculated into 100 mL of LB broth and incubated at 30 °C, 180 rpm, for 72 h. The fermentation solution was centrifuged for 15 min at 12,000 r/min, and the supernatant was filtered with a 0.22 μm bacterial filter to obtain the sterile filtrate. The bacteria-free filtrates of MM were mixed with the FON conidial suspension, and final dilutions of the sterile filtrate were prepared at 2, 10, 20, 40, 100, 200, and 300 times, respectively. The FON conidial suspension was mixed with the LB broth served as the control. The mixtures were added to sterile concave slides and cultured at 28 °C. After 24 h, spore germination was observed using a microscope (Nikon Ei, Nikon Precision Machinery (Shanghai) Co., Ltd., Shanghai, China), and 80–100 spores were examined per visual field. Each treatment was replicated three times. The rate of spore germination inhibition was calculated using the following formula: Spore germination inhibition rate (%) = [(A1 − A2)/A1 × 100], where A1 represents the spore germination rate of FON in the control, and A2 represents the spore germination rate of FON in the treatment.

To observe the effect of antagonistic strain MM on the mycelial morphology of FON, the hyphae from the edge of the bacteriostatic zone of an FON colony treated with *S. plymuthica* MM were collected. The hyphae of FON treated without strain MM were designated as the control. The collected mycelia were processed following the method described by Zhao et al. [23] and examined using scanning electron microscopy (ZEISS GeminiSEM 500, Oberkochen City, Germany).

### 2.4. Detection of Growth-Promoting and Biological Control Traits

The MM strain was cultured in LB broth at 180 rpm and 30 °C for 24 h. The bacterial cells were collected by centrifugation at 5000 rpm for 15 min at room temperature, and the bacteria were then resuspended in sterile water to a concentration of $1 \times 10^8$ CFU/mL for subsequent assays.

The production of extracellular enzymes by microorganisms is a crucial factor in the biocontrol of plant diseases. In this study, the activities of protease, chitinase, β-glucanase, and cellulase were detected to preliminarily explore the biocontrol mechanism of MM. The enzyme activities of enzymes were assayed by inoculating *S. plymuthica* MM on enzyme activity assay media [24,25]. The observation of a transparent zone around the colony

indicated that strain MM had the ability to secrete these four enzymes. Each treatment was performed with three replicates.

The production of growth-promoting substances by MM, including IAA and siderophores, was evaluated according to a modified method [26,27]. The production of siderophores was determined by the chromatic transition from blue to orange around the bacterial colony on a Chrome azurol S agar plate. IAA production was detected by growing MM cultures in L-tryptophan (100 mg/L) and without L-tryptophan King's B broth at 28 °C, 180 rpm, for 2 days. The amount of IAA produced by MM was calculated by comparing with the IAA standard curve.

### 2.5. Growth-Promoting Effect of MM on Watermelon Seedlings

The watermelon variety Jingxin No. 1, which is susceptible to FON, was used as the test plant in all pot experiments. The watermelon seeds were sterilized by treating them with 75% ethanol for 30 s, followed by rinsing them five times with sterile water. After drying, the seeds were sown in plastic flowerpots filled with sterilized soil and placed in a climate chamber at 25 °C with a photoperiod of 14 h light and 10 h darkness. Once the true leaves emerged, seedlings of similar size were selected and subjected to the following treatments: (1) roots drenched with 20 mL of FON conidial suspension ($10^5$ conidia/mL); (2) roots initially drenched with 20 mL of FON conidial suspension and, 24 h later, drenched with 20 mL of MM suspension ($10^8$ CFU/mL); (3) roots drenched with 20 mL of MM suspension; and (4) roots drenched with 20 mL of sterile water as a control. Three plants were used for each treatment, and each treatment was replicated three times. After 10 days, the roots of treatments (4) and (1) were drenched with 20 mL of sterile water, while the roots of treatments (2) and (3) were drenched with 20 mL of MM suspension. At day 35, the height of the plant shoots, root length, fresh weight, and dry weight were measured. Additionally, the chlorophyll content of watermelon leaves was determined using a chlorophyll extraction kit (Solarbio, Beijing, China) [24].

### 2.6. Colonization of MM in Watermelon Root

To observe the colonization of strain MM in watermelon roots, the GFP-marked vector (pMP2444) was utilized to construct GFP-tagged MM, following the methods described by Xiong et al. [28]. The GFP-MM was prepared as a bacterial suspension ($10^8$ CFU/mL) using the method outlined in Section 2.5. Two-leaf watermelon seedlings were transplanted into pots containing sterilized soil. The roots of the treatment group were drenched with 20 mL of GFP-MM cultures, while the control group received 20 mL of water. Each treatment was replicated in ten pots. The plants were cultivated in an artificial climate chamber at 25 °C, with a photoperiod of 14 h light and 10 h of darkness. At 5, 10, and 15 days after bacterial inoculation, the watermelon root samples were randomly collected for analysis. The collected roots were rinsed with water and cut into approximately 1 mm thick slices for examination using a confocal laser scanning microscopy (CLSM Olympus FV 3000, Olympus, Tokyo, Japan) with an excitation wavelength of 488 nm [29].

### 2.7. Effect of MM on Defense Enzyme Activity and Defense-Related Genes in Watermelon Seedlings

The watermelon seedlings were treated in the same manner as described in Section 2.5. After a period of 35 days, the leaves, stems, and roots from each treatment were collected and stored at −80 °C for further analysis. The activities of POD, PPO, CAT, and PAL were determined using assay kits following the manufacturer's instructions (Solarbio, Beijing, China) [24,30]. Absorbance readings were taken using a spectrophotometer (UA-2800A, UNIC, Shanghai, China).

In this study, five defense-related genes (*LOX*, *POD*, *PAL*, *C4H*, and *ClPR3*) were selected to evaluate the effect of *S. plymuthica* MM on the expression of watermelon resistance genes associated with activated defense mechanisms in plants. The roots of watermelon seedlings subjected to the same treatments as described in Section 2.5 were collected and stored at −80 °C for subsequent analysis. Total RNA was extracted following the instruc-

tions of the RNA extraction kit (SangonBiotec, Shanghai, China). The extracted RNA was then reverse-transcribed into cDNA using a reverse-transcription kit (TAKARA, Tokyo, Japan), and the resulting cDNA was stored at $-80$ °C.

The expression levels of *LOX*, *POD*, *PAL*, *ClPR3*, and *C4H* genes in watermelon roots treated with FON, FON + MM, and MM were determined by qRT-PCR. The qRT-PCR reactions were performed using a QuantStudioTM 6 Flex Real-Time PCR System (Mosaic, Hercules, CA) and a SYBR-Green kit (TAKARA, Tokyo, Japan). The qRT-PCR primers were designed by Sangon Biotech (Shanghai, China) (Table S2). 18S rRNA was used as the reference gene. The reaction was repeated three times, and the $2^{-\Delta\Delta CT}$ method was employed to calculate the relative expression levels of the genes [5].

### 2.8. Identification of Active Components of MM by GC-MS

To identify the antifungal substance of strain MM, a 72 h fermentation broth was centrifuged at 12,000 rpm for 15 min, and the supernatant was collected. The supernatant was then mixed with n-butanol, ethyl acetate, dichloromethane, and chloroform in a ratio of 1:1 ($v/v$) and separated using a separation funnel. The extract was concentrated using a rotary evaporator. The antifungal activity of the crude extracts obtained from different solvents was determined using the growth rate method. The inhibitory rate was used to express the antagonistic activity, and the extract with the highest inhibitory rate was selected for analysis using a GC-MS 7000C system (Agilent, Palo Alto, CA, USA). Each treatment was replicated three times. The test was completed in the laboratory of Lanzhou Jiaotong University Testing Center. The analysis conditions of the GC-MS were the same as reported by Abdelkhalek et al. [31]. The constituents were identified by comparing them with the available data in the GC-MS library.

### 2.9. Control Effect of MM on Watermelon Fusarium wilt in the Pot Experiment

In this experiment, the commercial chemical fungicide, hymexazol (15% aqueous solution), produced by Guangxi Tianyuan Biochemistry Co., Ltd., was used as a positive control for comparison with strain MM. Watermelon seedlings with two leaves were prepared following the method described in Section 2.5. Seedlings of the same size were subjected to the following treatments: (1) roots initially drenched with 20 mL of FON conidial suspension ($10^5$ conidia/mL) and then drenched with 20 mL of sterile water 24 h later; (2) roots drenched with 20 mL of FON conidial suspension and then drenched with 20 mL of MM suspension ($10^8$ CFU/mL) 24 h later; and (3) roots drenched with 20 mL of FON conidial suspension and then drenched with 20 mL of a 15% hymexazol aqueous solution (500 times) 24 h later. Each treatment consisted of three plants and was replicated three times. After 10 days, the roots of treatment (1) were drenched with 20 mL of sterile water, while the roots of treatments (2) and (3) were drenched with 20 mL of MM suspension and hymexazol aqueous solution, respectively. At day 42, the severity of Fusarium wilt was assessed using a 0 to 4 scale [24,32], and the disease index and control efficacy were calculated using the following two formulas: disease index (%) = $\sum(A \times B)/(C \times 4) \times 100$, where A was the number of diseased plants in each grade, B was the disease grade, and C was the total number of plants assessed; control efficacy (%) = $[(F1 - F2)/F1] \times 100$, where F1 was the disease index of the control, and F2 was the disease index of the treatment.

### 2.10. Statistical Analyses

The experimental data were analyzed using SPSS 24.0 software and expressed as the mean $\pm$ SD. The mean values were analyzed using one-way analysis of variance (ANOVA), followed by Duncan's multiple range test at the 5% ($p < 0.05$) level of significance.

## 3. Results

### 3.1. Isolation, Screening, and Identification of Antagonistic Bacteria against FON

In this study, a total of 68 bacterial strains were isolated from the 10 herbage rhizosphere soil samples and used for the antagonism test, and eight bacteria with varying

degrees of antagonistic activity against FON were obtained. Among them, strain MM exhibited the highest inhibitory activity against FON, with an inhibition rate of 65.46% (Figure 1). Due to its strong inhibitory effect, strain MM was selected for further investigation.

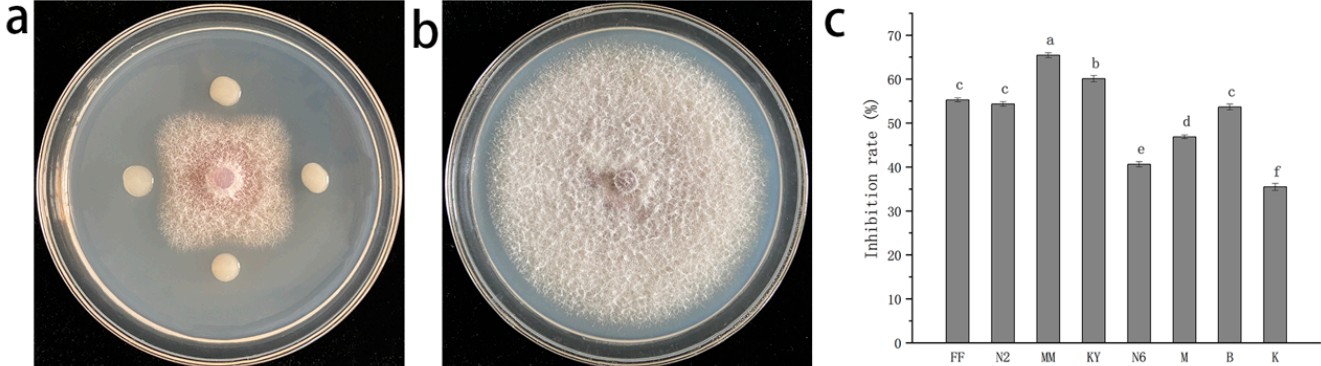

**Figure 1.** Antagonistic effect of MM against FON. (**a**) FON treated with MM: MM was inoculated on the plates approximately 2.5 cm away from the plug of FON at four points, and the plates showed obvious inhibition zones. (**b**) Control: PDA plates without inoculation of MM. (**c**) Inhibition rate of eight bacterial strains against FON. Experiments were repeated three times, and error bars are the means $\pm$ SD for treatment (N = 3). Different lowercase letters indicate significant difference at $p < 0.05$ level by Duncan's new multiple range test.

Morphological observation of strain MM under a microscope revealed that it was Gram-negative and appeared as short rod-shaped cells. When cultured on LB medium at 28 °C for 3 days, the colony of strain MM had a well-defined edge, a smooth surface, and a milky white color (Figure S1). Phylogenetic analysis based on the 16S rDNA sequence confirmed that the strain MM and *Serratia plymuthica* clustered together (Figure 2). Therefore, the strain MM was identified as *Serratia plymuthica*.

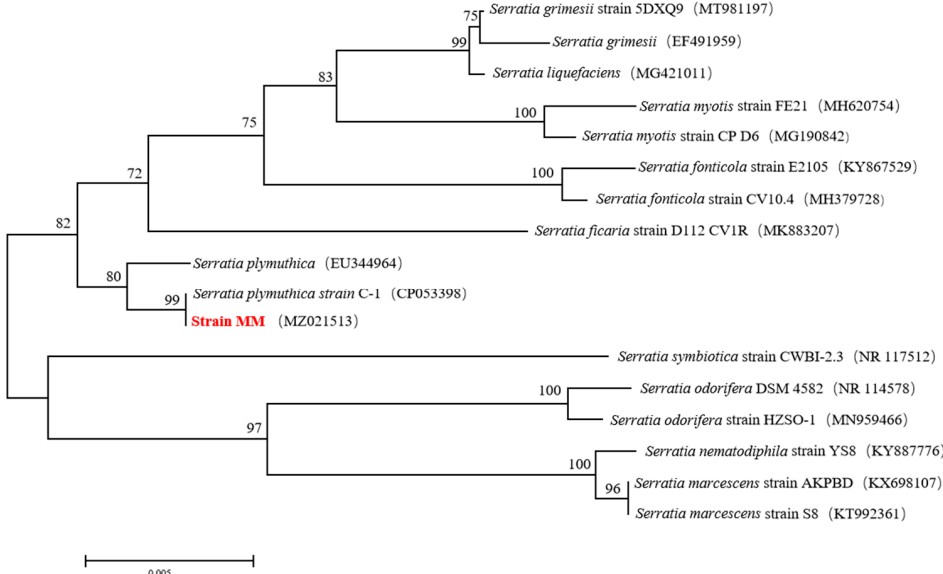

**Figure 2.** Phylogenetic tree based on the 16S rDNA sequence. Phylogenetic tree was constructed by the neighbor-joining method of MEGA 6.0 with bootstrap values based on 1000 replications. The scale bar represents the number of substitutions per base position.

### 3.2. Inhibition Spectrum of S. plymuthica MM

To determine the antifungal spectrum, its antagonistic activity of strain MM against eight other plant pathogenic fungi was assessed. The results revealed that *S. plymuthica*

MM exhibited inhibitory effects on the growth of all tested fungal strains. Among them, the strongest inhibitory activity was observed against *Botrytis cinerea*, with an inhibitory rate of 74.00%. The inhibitory rates against other plant pathogenic fungi ranged from 60–70% (Figure 3, Table 1). These findings highlight the broad-spectrum antifungal activity of strain MM, particularly against soil-borne diseases caused by Fusarium pathogens.

**Table 1.** Inhibitory activity of *S. plymuthica* MM against eight plant pathogenic fungi.

| Phytopathogen | Plant Disease | Treatment Colony Diameter (cm) | Control Colony Diameter (cm) | Inhibition Ratio (%) |
|---|---|---|---|---|
| *Botrytis cinerea* | Gray mold of strawberry | 2.04 ± 0.075 | 7.83 ± 0.019 | 74.00 a |
| *Fusarium graminearum* | Wheat scab | 3.28 ± 0.029 | 8.41 ± 0.052 | 60.96 c |
| *Colletotrichum gloeosporioides penz* | *Lycium barbarum* Anthrax | 2.86 ± 0.126 | 8.05 ± 0.018 | 64.38 b |
| *Fusarium proliferatum* | Rice panicle rot | 3.17 ± 0.078 | 8.51 ± 0.003 | 62.69 c |
| *Fusarium oxysporum* | Fusarium wilt of cotton | 2.64 ± 0.039 | 8.16 ± 0.021 | 67.64 b |
| *Fusarium moniliforme* | Fusarium wilt of lily | 2.82 ± 0.058 | 8.08 ± 0.008 | 65.13 b |
| *Fusarium solani* | Root rot of *Astragalus membranaceus* | 2.73 ± 0.029 | 8.51 ± 0.019 | 67.90 b |
| *Fusarium avenaceum* | Root rot of *Angelica sinensis* | 3.07 ± 0.029 | 8.52 ± 0.005 | 63.99 bc |

Note: The mean and standard deviation (SD) of the data are shown (N = 3); different letters on the same line denote significant differences at the 0.05 level of *p*-value by Duncan's new multiple range test.

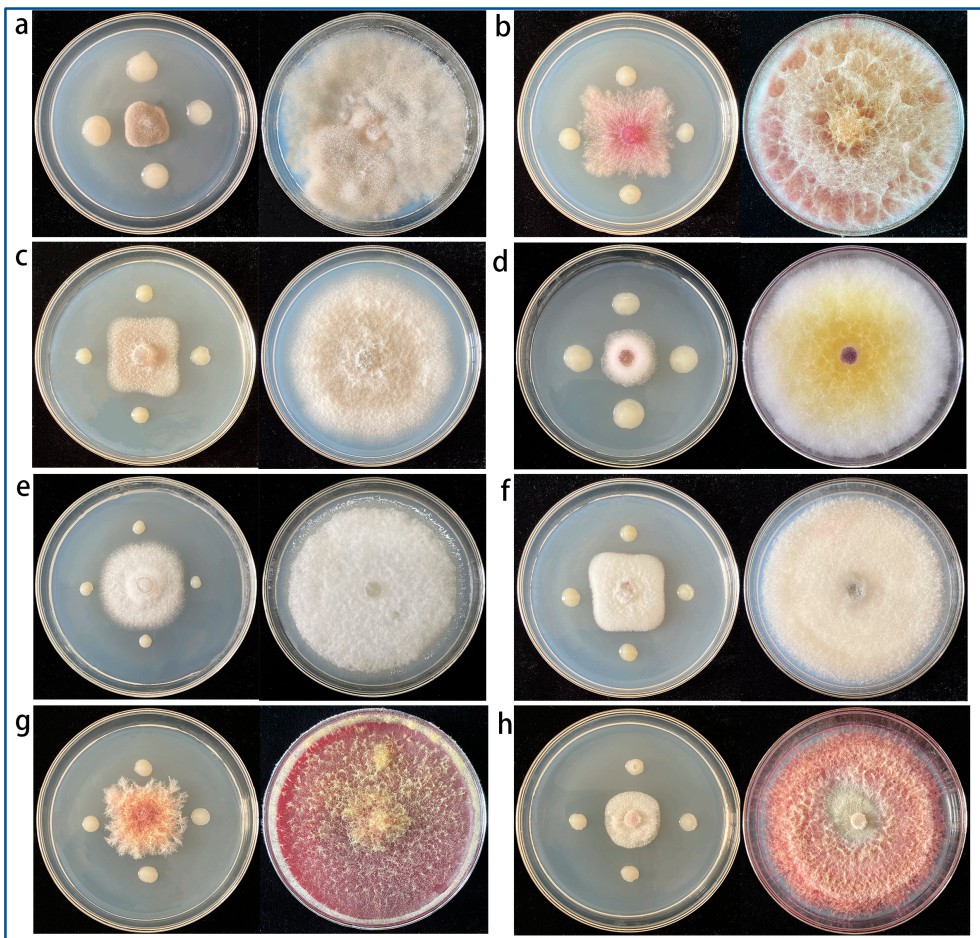

**Figure 3.** Antagonistic effects of *S. plymuthica* MM on eight phytopathogens. The MM was inoculated on the plates approximately 2.5 cm away from the plug of different fungi at four points. (**a**) *B. cinerea*; (**b**) *F. graminearum*; (**c**) *C. gloeosporioides penz*; (**d**) *F. proliferatum*; (**e**) *F. oxysporum*; (**f**) *F. moniliforme*; (**g**) *F. solani*; (**h**) *F. avenaceum*.

*3.3. Effect of S. plymuthica MM on Mycelial Morphology and Spore Germination of FON*

To observe the inhibitory effect of strain MM on the pathogenic fungi FON, the hyphae of FON treated with MM were examined using an electron microscope, and the

spore germination was observed under a microscope. The SEM analysis revealed that the morphology of FON mycelia treated with MM exhibited significant collapse, shrinkage, and deformity (Figure 4b,c). In contrast, the mycelia in the control group appeared regular and smooth (Figure 4a).

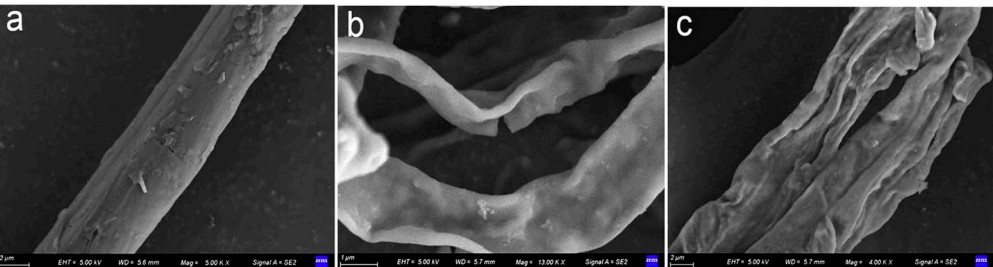

**Figure 4.** SEM observations of the morphology of FON treated with *S. plymuthica* MM. (**a**) Control: the hyphae of FON treated without strain MM; the hyphae surface was smooth. (**b,c**) The hyphae of FON treated by *S. plymuthica* MM for 5 days; the hyphae showed shrinkage and collapse.

The results of the FON spore germination inhibition test demonstrated that the sterile filtrate of MM was capable of inhibiting spore germination at a different dilutions (2, 10, 20, 100, 200, and 300 times), with the inhibition rates decreasing as the dilution increased. Particularly, at dilutions of 2 and 10, the MM filtrate exhibited remarkable inhibition effects, with inhibition rates of 89.68% and 84.89%, respectively (Figure S2). These findings collectively indicate that MM can disrupt the mycelial structure and inhibit spore germination, effectively impeding the development of FON.

### 3.4. Estimation of Plant Growth-Promotion Substance and Extracellular Enzymes in MM

To assess the biological control-related traits of strain MM, the activities of four enzymes were tested on skim milk agar culture, colloidal chitinase medium agar culture, β-glucanase agar culture, and sodium carboxymethylcellulose agar culture, respectively. As depicted in Figure 5, the presence of a transparent zone around the colony was observed on the enzyme activity assay medium indicated that strain MM possesses the ability to secrete protease (Figure 5a), chitinase (Figure 5b), β-glucanase (Figure 5c), and cellulase (Figure 5d). The production of these enzymes is considered one of the factors contributing to the biological control traits of MM.

Furthermore, the production of IAA and siderophores by strain MM was determined using the Salkowski colorimetric assay and O-CAS assay, respectively. The results revealed that MM was capable of producing siderophores (Figure 5e) and IAA (Figure 5f). The amount of IAA produced in King's B medium containing L-tryptophan was measured to be 10.239 mg/L, whereas in King's B medium without L-tryptophan, it was 5.14 mg/L. These results suggest that the presence of L-tryptophan enhances the capability of strain MM to produce IAA. Overall, these findings indicate that MM has the ability to produce plant-growth-promoting substances.

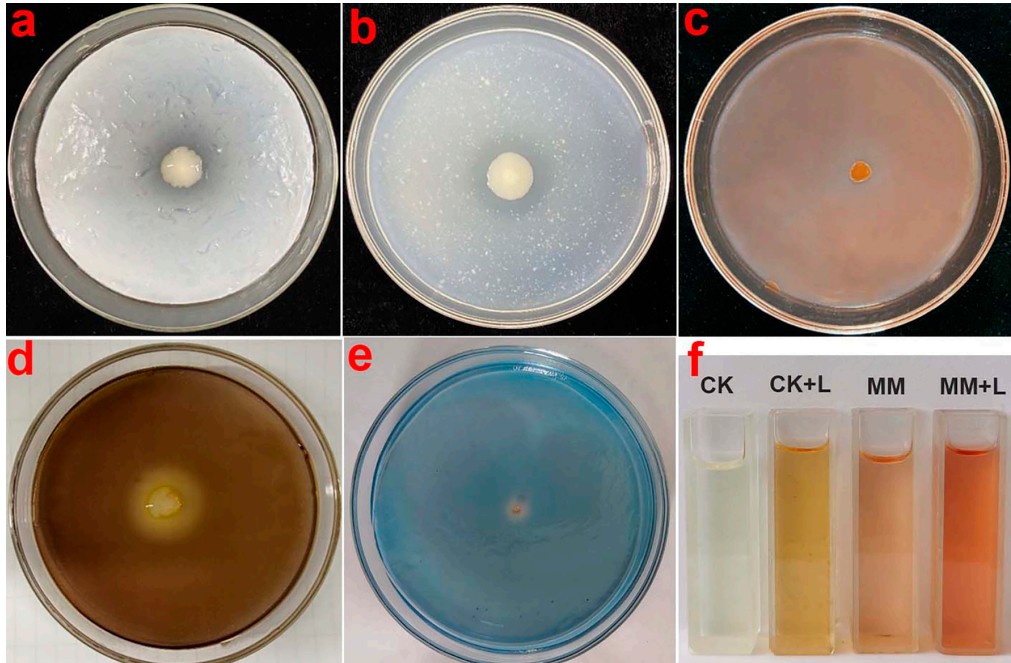

**Figure 5.** Detection of extracellular enzymes and growth promotion substances. The activity test of protease (**a**), chitinase (**b**), β-glucanase (**c**), and cellulase (**d**) produced by *S. plymuthica* MM; a transparent zone was observed around the colonies on the plates. Siderophores detection of MM-produced siderophore, and halos were observed around the colonies on the O-CAS assay (**e**). The IAA produced by MM in the Kim's medium containing L-tryptophan and without L-tryptophan (**f**). CK, treatment with sterile water; CK + L, treatment with sterile water and L-tryptophan; MM, treatment with MM; MM + L, treatment with MM and L-tryptophan.

### 3.5. Colonization of S. plymuthica MM in Watermelon Root

The colonization of beneficial microorganisms in plant roots plays a crucial role in their beneficial effects. In this study, the colonization of MM in watermelon roots was visualized by examining the fluorescent cells of GFP-MM using CLSM at different time points. Five days after inoculation, GFP-MM was observed on the surface layer of the watermelon root, exhibiting the highest fluorescence intensity (Figure 6a). Furthermore, GFP-tagged MM cells were observed to penetrate the interior of the watermelon root after 10 days of inoculation (Figure 6b). By day 15, the fluorescence intensity in the roots decreased and became weak. This observation indicates that *S. plymuthica* MM can effectively colonize the roots of watermelon.

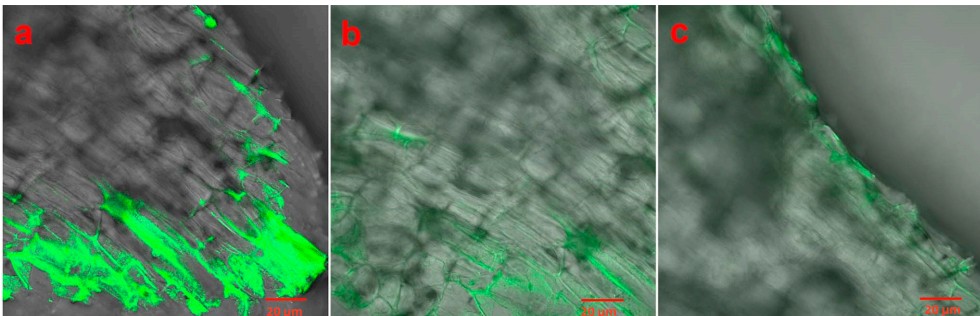

**Figure 6.** The observation of fluorescent cells of GFP-MM under CLSM at 5 (**a**), 10 (**b**), and 15 (**c**) days after bacterial inoculation.

### 3.6. Growth-Promotion Activity of S. plymuthica MM

After 35 days of inoculation, we measured the morphological indicators of watermelon seedlings treated with MM. Compared to the water treatment, the watermelon seedlings treated with MM showed significant improvements in plant height, root length, fresh weight, and dry weight, with increases of 78.83%, 62.29%, 126.90%, and 82%, respectively. Additionally, the content of chlorophyll a, chlorophyll b, and total chlorophyll in the leaves increased by 14.89%, 73.19%, and 35.39%, respectively (Figure 7).

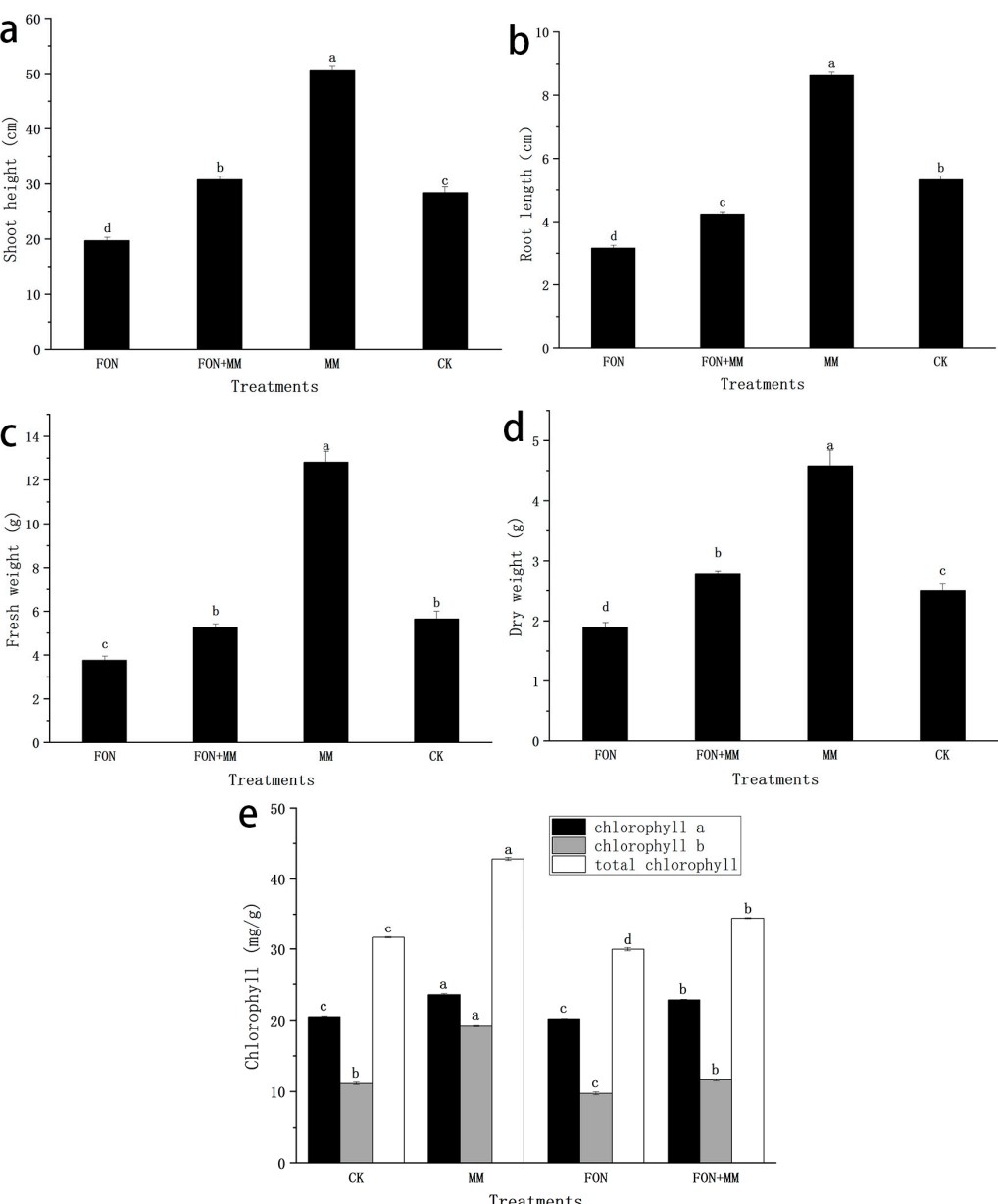

**Figure 7.** Effects of *S. plymuthica* MM on watermelon seedlings growth and the chlorophyll content. (**a**) Shoot height; (**b**) root length; (**c**) fresh weight; (**d**) dry weight. The watermelon plants at 35 days after treatment with sterile water (CK), *S. plymuthica* MM (MM), plant pathogen FON (FON), and plant pathogen FON and *S. plymuthica* MM (FON + MM). (**e**) Chlorophyll content of watermelon leaves at 35 days after treatment with sterile water (CK), *S. plymuthica* MM (MM), plant pathogen FON (FON), and plant pathogen FON and *S. plymuthica* MM (FON + MM). Three plants were used for each treatment, and each treatment was replicated three times with similar results. Note: Error bars are the means ± SD from for each treatment (N = 9), and different lowercase letters indicate significant difference at $p < 0.05$ level by Duncan's new multiple range test.

In the FON + MM treatment, there was also a significant increase in the content of chlorophyll a, chlorophyll b, and total chlorophyll as well as in shoot height, root length, fresh weight, and dry weight compared to the FON treatment (Figures 7 and 8). These results demonstrate that MM not only promotes the biomass of watermelon plants but also enhances the chlorophyll content in the leaves. This suggests that MM exhibits notable growth-promoting effects on watermelon seedlings regardless of whether the plants are infected with FON or not.

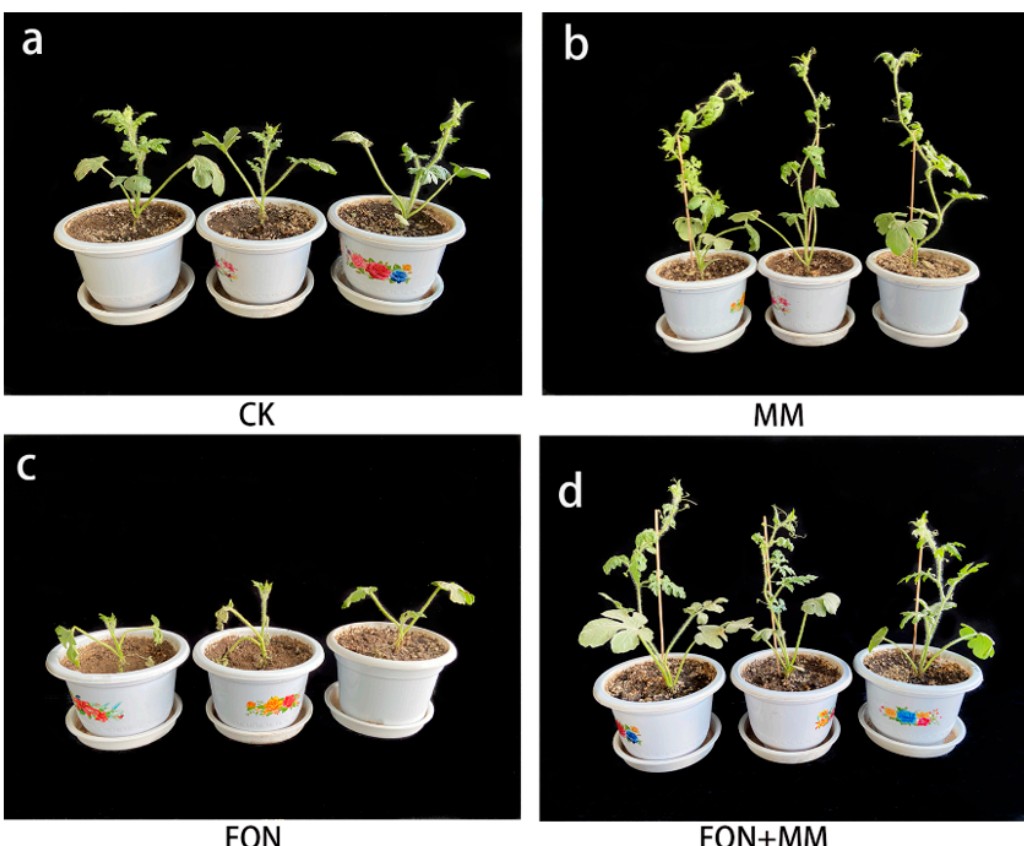

**Figure 8.** Growth-promotion effect of *S. plymuthica* MM on watermelon seedlings. Growth phenotype of watermelon plants at 35 days after treatment with sterile water (**a**), MM (**b**), FON (**c**), and FON + MM (**d**), respectively. Each treatment consists of nine plants.

### 3.7. Effect of S. plymuthica MM on the Activities of Defense-Related Enzyme

The resistance mechanism of plants to pathogens is closely related to the activities of defense-related enzyme. After 35 days of inoculation, we measured the activities of defense-related enzymes including POD, PPO, PAL, and CAT in the roots, stems, and leaves of the watermelon plants. Inoculation with strain MM led to a significant increase in the activities of these enzymes compared to the water treatment. In the FON treatment, the activities of POD, PPO, PAL, and CAT also increased compared to the control group; however, when compared to the MM treatment, the activities of POD, PPO, and PAL were lower except for CAT. In the FON + MM treatment, the activities of all four defense-related enzymes in the roots, stems, and leaves of watermelon seedlings were higher than in the FON treatment (Figure S3). These results indicate that strain MM can stimulate the production of defense enzymes in watermelon seedlings even when the plants are infected by FON. This suggests that MM enhances the plant's defense mechanisms against the pathogen and contributes to improved disease resistance.

### 3.8. Effect of S. plymuthica MM on Resistance Gene Expression in Watermelon Plants

In addition to assessing the effects of *S. plymuthica* on the activity of four defense-related enzymes, we also evaluated the relative expression levels of five defense-related genes to further understand the mechanism of *S. plymuthica* in controlling watermelon wilt disease. Compared to the control group (CK), the MM treatment showed significant up-regulation of the *LOX*, *POD*, *PAL*, *ClPR3*, and *C4H* genes (Figure 9). In the FON + MM treatment, only the expression of the POD gene was up-regulated compared to the FON treatment. These results indicate that strain MM can cause a noticeable up-regulation of the five defense-related genes when the watermelon roots are not infected with FON. However, when the roots are infected with FON, MM does not significantly increase the expression of these defense-related genes.

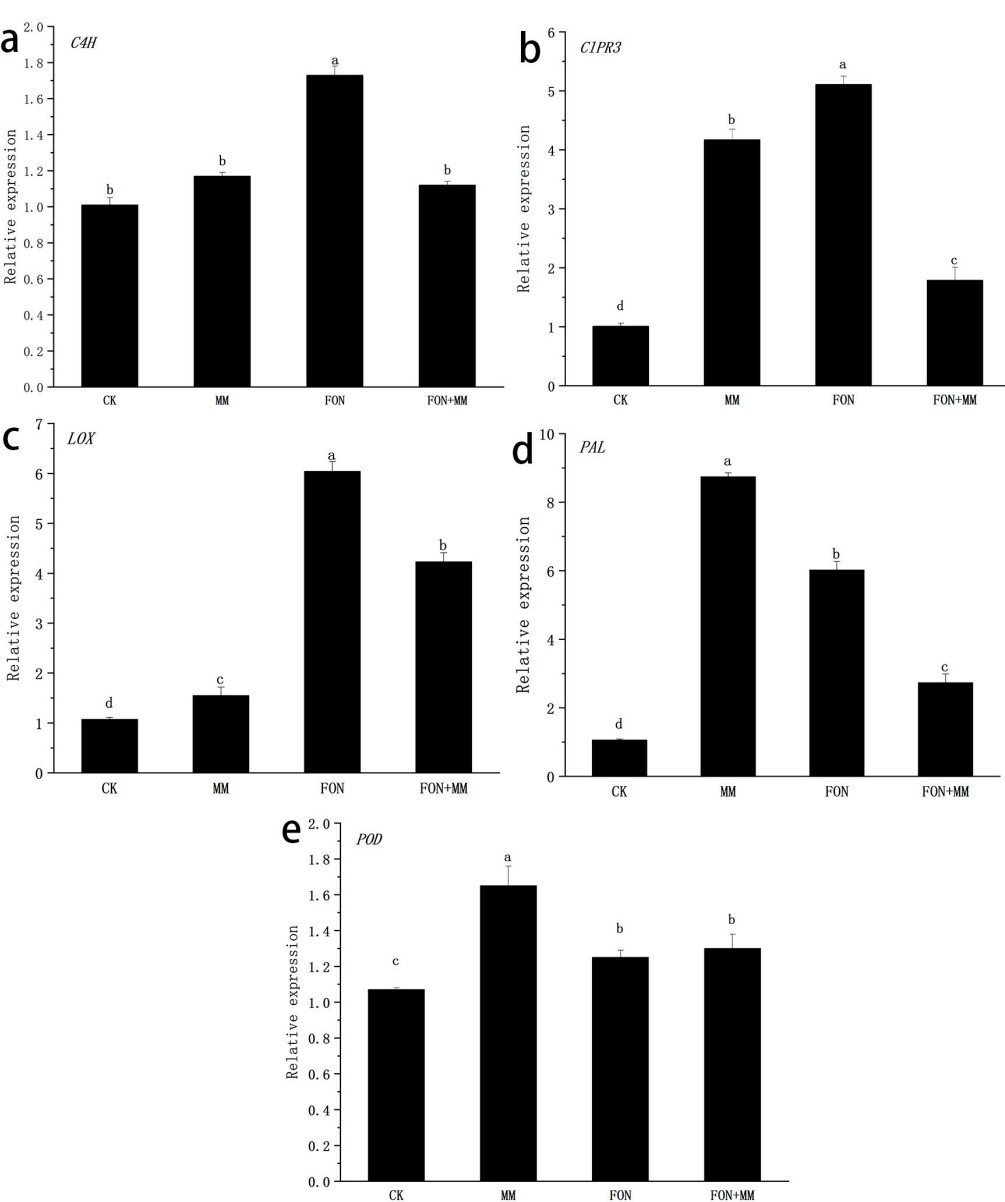

**Figure 9.** Effect of *S. plymuthica* MM on the defense-related genes expression. Relative expression of *C4H* gene (**a**), *ClPR3* gene (**b**), *LOX* gene (**c**), *PAL* gene (**d**), and *POD* gene (**e**) in watermelon roots treated with sterile water (CK), MM (MM), FON (FON), and FON and MM (FON + MM). Each treatment was replicated three times. Note: Error bars are the means ± SD from for each treatment (N = 3), and different lowercase letters indicate significant difference at $p < 0.05$ level by Duncan's new multiple range test.

### 3.9. Identification of Bioactive Substances of S. plymuthica MM

The antimicrobial activity of biocontrol bacteria is mainly attributed to secondary metabolites, which can directly inhibit the growth of plant pathogens and induce systematic resistance in plants to control plant diseases. In our study, we extracted the bioactive substances from the culture filtrate of *S. plymuthica* MM using n-butanol, ethyl acetate, dichloromethane, and trichloromethane. The results showed that the n-butanol extract exhibited the highest inhibitory effect on FON, with an inhibition rate of 85.81% (Figure 10). Therefore, the n-butanol extract was selected for further analysis using GC-MS.

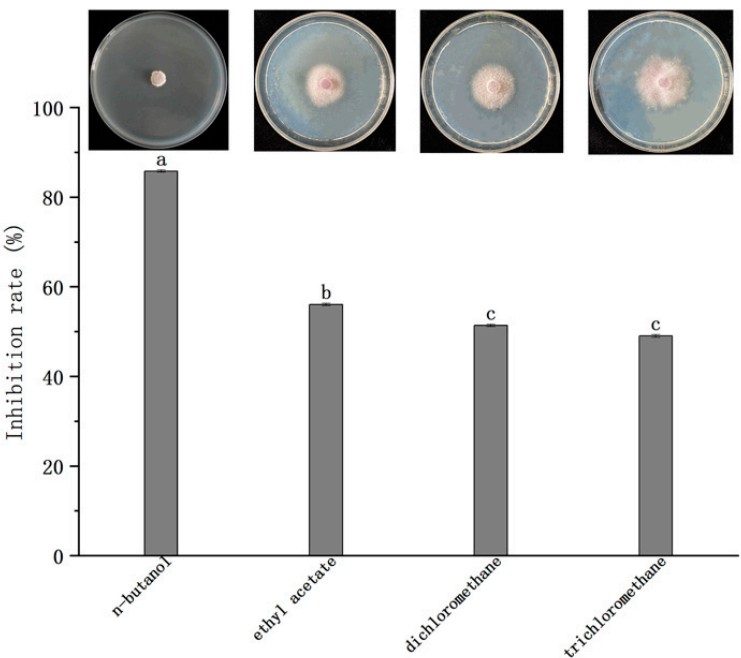

**Figure 10.** Antifungal activity of the extracts of different extractants against FON. Different lowercase letters indicate significant difference at $p < 0.05$ level by Duncan's new multiple range test.

The GC-MS analysis of the n-butanol extract revealed the presence of five bioactive substances. These compounds were identified as follows: 2,5-piperazinedione, 3-methyl-6-(1-methylethyl)-(0.42%); pyrrolo[1,2-a]pyrazine-1,4-dione,hexahydro-(2.98%); pyrrolo[1,2-a]pyrazine-1,4-dione, hexahydro-3-(2-methylpropyl)-(1.16%); 2,5-piperazinedione,3,6-bis(2-methylpropyl)-(1.07%); and 9-Octadecenamide,(Z)-(3.45%). The chemical properties and structures of these compounds are described in Table 2 and illustrated in Figure 11.

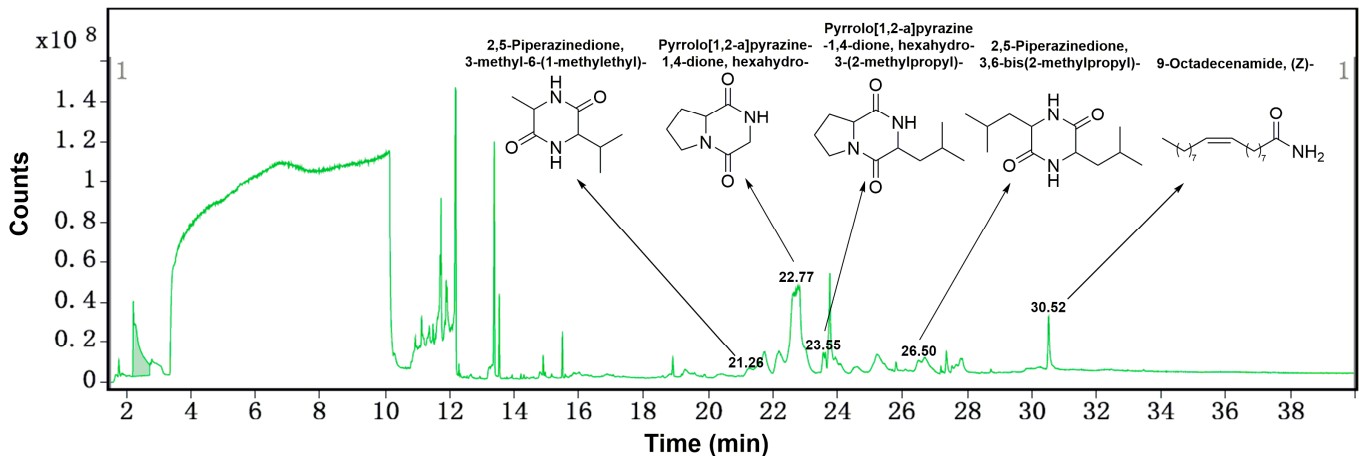

**Figure 11.** The chemical structures of five compounds from n-butanol extract of *S. plymuthica* culture filtrate.

**Table 2.** The chemical properties of five compounds from n-butanol extract of *S. plymuthica* culture filtrate.

| Retention Time (min) | Compounds | Area % | Chemical Formula | Molecular Weight (g/mol) |
|---|---|---|---|---|
| 21.26 | 2,5-Piperazinedione, 3-methyl-6-(1-methylethyl)- | 0.42 | $C_8H_{14}N_2O_2$ | 170.21 |
| 22.77 | Pyrrolo[1,2-a]pyrazine-1,4-dione, hexahydro- | 2.98 | $C_7H_{10}N_2O_2$ | 154.17 |
| 23.55 | Pyrrolo[1,2-a]pyrazine-1,4-dione,hexahydro-3-(2-methylpropyl)- | 1.16 | $C_{11}H_{18}N_2O_2$ | 210.28 |
| 26.50 | 2,5-Piperazinedione, 3,6-bis(2-methylpropyl)- | 1.07 | $C_{12}H_{22}N_2O_2$ | 226.32 |
| 30.52 | 9-Octadecenamide, (Z)- | 3.45 | $C_{18}H_{35}NO$ | 281.48 |

*3.10. Control effect of S. plymuthica MM on Watermelon Fusarium Wilt*

In the pot experiment, the watermelon plants that were only inoculated with FON displayed symptoms of wilt disease, including severe discoloration of vascular bundles and withering of some leaves. The disease index for this group was recorded as 90. However, the watermelon plants treated with *S. plymuthica* MM and hymexazol aqueous showed only slight discoloration of the vascular bundles, indicating a lower severity of wilt disease. The control efficacy of *S. plymuthica* MM and hymexazol aqueous treatments was measured to be 74.07% and 75.92%, respectively. There was no significant difference in control efficacy between these two treatments (Table 3, Figure S4). These results demonstrate that strain MM exhibited good control effectiveness against Fusarium wilt of watermelon.

**Table 3.** Control effect of *S. plymuthica* MM against Fusarium wilt of watermelon.

| Treatment | Disease Index | Control Efficacy (%) |
|---|---|---|
| FON | $90.00 \pm 0.08$ a | - |
| FON + MM | $23.33 \pm 0.02$ b | 74.07 a |
| FON + Hymexazol | $21.67 \pm 0.04$ b | 75.92 a |

Note: The mean and standard deviation (SD) of the data are shown (N = 9); different letters on the same line denote significant differences at the 0.05 level of *p*-value.

## 4. Discussion

Fusarium wilt is a major threat to watermelon production, leading to yield reduction or complete harvest failure [4]. Biological control offers a promising approach for managing fungal diseases, providing an alternative to chemical fungicides. The use of biocontrol agents offers several advantages, including pathogen reduction, environmental friendliness, promotion of plant growth, enrichment of beneficial microbial communities, improved crop resistance, enhanced soil fertility, increased fertilizer efficiency, and mitigation of cultivation obstacles [11,33–35]. Therefore, continuous screening and discovery of highly efficient and long-lasting beneficial microorganisms for soil-borne disease control is necessary. Additionally, further research on the growth-promoting and biocontrol properties of these microorganisms will contribute to their development as microbial pesticides or fertilizers.

In this study, *S. plymuthica* MM was isolated as a biocontrol bacterium with strong antagonistic activity against FON. MM demonstrated the ability to inhibit FON mycelial growth and spore germination in vitro. The pot experiment demonstrated that MM exhibited comparable control efficacy to the chemical fungicide hymexazol against watermelon Fusarium wilt. These results highlight the potential of MM for development as a biocontrol agent for managing Fusarium wilt in watermelon. Previous studies have reported significant control of watermelon Fusarium wilt using *Bacillus velezensis* F21, *Fluorescent pseudomonads* WMC16-1-1, and *Bacillus subtilis* IBFCBF-4 [36–38]. However, the use of *S. plymuthica* for watermelon wilt control has not been reported until now, making this study the first to propose *S. plymuthica* as a biological control agent for watermelon Fusarium wilt. Furthermore, in vitro antagonism tests revealed that strain MM exhibited a broad antifungal spectrum, which is consistent with preceding reports [14–16].

Effective colonization is a crucial factor for promoting plant growth, facilitating plant–microorganism interactions, and inhibiting the spread of phytopathogens [39]. In our study, GFP-tagged MM was observed to colonize the watermelon roots both on the surface and internally. This finding indicates that MM can establish a long-lasting colonization in watermelon roots, providing protection against pathogen infections and promoting plant growth. This colonization pattern is consistent with studies on *Paenibacillus polymyxa* and *Bacillus tequilensis* in *Angelica* root [29]. However, it should be noted that the population size of GFP-MM colonized in the watermelon roots was not quantified in this study, and the long-term stability of strain MM in watermelon roots cannot be confirmed based on these findings. Therefore, in the future, the population number of GFP-MM colonized in watermelon will be counted.

In the pot experiment, the antagonistic bacterium MM promoted the growth of watermelon seedlings even in the presence of FON infection. The watermelon treated with antagonistic bacteria resulted in higher agronomic characters than control. This is similar to the effects of *Bacillus subtilis* YB-04, which controls cucumber wilt and enhances cucumber seedling growth [24]. Biocontrol bacteria can promote plant growth through the production of various substances [40]. IAA, as a key plant hormone, plays a significant role in regulating plant growth and enhancing plant resistance [27,41]. *S. plymuthica* MM has the ability to produce IAA, and the addition of L-tryptophan further enhances IAA production. Additionally, *S. plymuthica* MM produces siderophores, which improve iron absorption by roots and alleviate the detrimental effects of iron on crops [42]. Therefore, *S. plymuthica* MM promotes the growth of watermelon seedlings through the production of siderophores and IAA.

Numerous studies have demonstrated the crucial role of hydrolytic enzymes in plant disease control [24,43]. In our study, *S. plymuthica* MM exhibited the ability to produce protease, amylase, cellulase, and β-glucanase. These enzymes can degrade polymers such as chitin and glucan in the fungal cell wall, resulting in cell structure deformation and leakage of cell contents, thereby impeding the spread of fungal pathogens [44,45]. This may explain the observed deformation of FON mycelium. Furthermore, our study revealed that strain MM could activate defense mechanisms in watermelon. The resistance mechanism is related to the activity of defense-related enzymes. Defense-related enzymes, including POD, PPO, CAT, and PAL, play a significant role in lignin synthesis, detoxification of high-concentration free radicals, regulation of reactive oxygen species (ROS) homeostasis, production of toxic compounds for direct pathogen elimination, and enhancement of plant resistance against fungal pathogens [46,47]. In our research, whether the plants were infected by FON or not, the activities of POD, PPO, CAT, and PAL significantly increased in the watermelon seedlings inoculated with *S. plymuthica* MM. Thus, the control effect against Fusarium wilt of watermelon may be due to the ability of antagonistic bacterium MM to secrete hydrolytic enzymes and induce plants to secrete resistance enzymes.

Plant-induced resistance is closely associated with the jasmonic acid synthesis pathway (JA), shikimic acid phenylpropanoid–lignin synthesis pathway, and salicylic acid pathway (SA) [48]. Previous studies have reported that induced gene expression of SA, JA, and styrene–acrylate lignin shikimic acid biosynthesis pathways is beneficial for stimulating the plant's immune system against FON [49–51]. In our study, the *LOX* gene is associated with JA, the *ClPR3* gene is associated with SA, and the *POD*, *PAL*, and *C4H* genes are associated with the phenylpropanoid–lignin synthesis pathway. The data indicated that MM induced defense-related gene expression levels in watermelon roots in comparison with the control. This finding is consistent with previous reports on *Bacillus subtilis* MBI600, *Bacillus cereus* EC9, and *Bacillus amyloliquefaciens*, which trigger the plant's immune system against fungal phytopathogens by inducing some biosynthetic pathways in the plant [9,52,53]. Furthermore, with the inoculation of FON, the expression of defense-related genes including *C4H*, *ClPR3*, and *LOX* were enhanced. This is consistent with findings of previous works suggesting that PGPR triggers only mild defense responses compared to those triggered by plant pathogens [9]. This study showed that the MM does not significantly increase the expression of the five defense-related genes when the roots are infected with FON. However,

some previous studies suggesting that combined application of antagonistic bacterium and artificial inoculation with plant pathogen resulted in an increase of the defense-related gene expression [9,54]. This may be caused by a different inoculation sequence. In this study, the pathogen FON was inoculated first, and biocontrol bacterium MM was inoculated subsequently, which was different from the inoculation sequence in previous studies. This is because plant resistance to pathogens after exposure to beneficial microbes is associated with its priming effects, and priming has been associated with the production of metabolites using PGPR species [9]. But inoculation of FON before inoculation of MM may not bring out this priming effect of biocontrol bacteria MM.

Beneficial microorganisms typically produce secondary metabolites with biological activities that can induce plant defense responses or directly inhibit the reproduction of phytopathogens [55,56]. In our study, the analysis revealed that five active metabolites were produced by MM. Based on recent research, pyrrolo[1,2-a]pyrazine-1,4-dione, hexahydro- and pyrrolo[1,2-a]pyrazine-1,4-dione, hexahydro-3-(2-methylpropyl)- are pyrrolizines compounds that have demonstrated good antifungal effects against *Fusarium oxysporum*, *Fusarium oxysporum* f. sp. *Lycopersici*, and *Pyricularia oryzae*. These compounds inhibit the synthesis of functional proteins by down-regulating specific functional protein coding genes [31,57,58]. 2,5-Piperazinedione, 3-methyl-6-(1-methylethyl)- and 2,5-Piperazinedione, 3,6-bis(2-methylpropyl)- are found in many strains of *Bacillus* and have been reported to exhibit strong antifungal activity against various phytopathogenic fungi [31,59–62]. Additionally, 9-octadecenamide, (Z) has also been reported to possess antimicrobial properties [63,64]. In conclusion, MM exhibits antifungal activity against FON through the production of these active substances.

## 5. Conclusions

In conclusion, the isolated *S. plymuthica* MM demonstrated effective control of watermelon Fusarium wilt. It exhibited a broad antifungal spectrum, inhibiting the growth of various plant-pathogenic fungi. Strain MM produced extracellular enzymes and five bioactive substances that contributed to the inhibition of FON growth. Furthermore, MM exhibited the ability to colonize watermelon roots and enhance plant resistance against FON through the stimulation of defense enzymes and up-regulation of defense-related genes. In pot experiments, MM also showed growth-promoting activity in watermelon plants. This study represents the first instance of *S. plymuthica* being used as a biocontrol agent against watermelon Fusarium wilt, providing a foundation for the development of a novel biocontrol approach for this disease.

**Supplementary Materials:** The following supporting information can be downloaded at: https://www.mdpi.com/article/10.3390/agronomy13092437/s1, Figure S1: Morphological of *Serratia plymuthica* MM; Figure S2: Effect of culture filtrates of *Serratia plymuthica* MM on FON spore germination; Figure S3: Effect of strain MM on the resistance enzyme activities of watermelon roots, stems, and leaves; Figure S4: Control effect of strain MM on watermelon Fusarium wilt in the greenhouse; Table S1: The 16S rDNA sequence of universal primers; Table S2: The primers of defense-related genes used in qRT-PCR.

**Author Contributions:** Conceptualization, T.T., Z.X., Z.L., J.M. and Y.T.; methodology, Z.L., J.L., X.S. and J.M.; validation, J.M. and J.L.; formal analysis, T.T., Z.L., Z.X., J.M. and J.L.; investigation, Z.L. and T.S.; writing—original draft preparation, Z.L., J.M. and J.L.; writing—review and editing, Z.L., Z.X., J.L., Y.C., T.T., J.M. and Y.T. All authors have read and agreed to the published version of the manuscript.

**Funding:** This research was funded by Gansu Jiayuguan city science and technology key research and development project (21-19); Key R&D program of Shandong Province (2021CXGC010804); Industrial Support Program of Gansu Province Education Department (2022CYZC-39); Qinghai Provincial Central Government Guide Local Science and Technology Development Project (2023ZY019); Key Research and Development Program of Gansu Province (21YF5FA059).

**Data Availability Statement:** Not applicable.

**Conflicts of Interest:** The authors declare no conflict of interest.

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
