# Peer review of "A Biocontrol Strain of Serratia plymuthica MM Promotes Growth and Controls Fusarium Wilt in Watermelon"

_agronomy, doi:10.3390/agronomy13092437_

Round 1

Reviewer 1 Report

In this submission, the authors report the isolation, identification and characterization of a Serratia plymuthica strain (MM) from soil that possesses antifungal activity. It was tested against Fusarium oxysporum f. sp. niveum (FON), the causal agent of Fusarium wilt in melon. The bacterium can colonize melon plant roots, promoted their growth and provided protection against FON. Several antagonism mechanisms were explored such as production of extracellular enzymes and two antifungal compounds (Piperazinedione, Pyrrolo [1,2-a]pyrazine-1,4-dione and Octade- 33cenamide).

The materials and methods section provides great detail on the experiments and valuable data on the characterization of the Serratia plymuthica strain (MM) is provided as well as its effects on the melon plants. The manuscript is too long, in particular the Results and Discussion sections. Some results could be moved to supplementary material.

The manuscript contains some conceptual errors that are concerning. For example:

In the abstract (Lines 22 – 23): “Additionally, S. plymuthica MM demonstrated antagonistic activity against eight plant pathogens, indicating that MM had broad-spectrum antibacterial activity”. The eight plant pathogens tested were all fungi.

Line 290: “To observe the inhibitory effect of strain MM on the pathogenic bacterium FON, the hyphae of FON treated with MM were examined”. FON is a fungi.

Other comments:

The last paragraph in the introduction (Lines 73 – 81) is a short summary of the results. Instead, it could be replaced by information on the novelty of this work.

Line 92: Why this watermelon variety was selected. How does it fare against F. oxysporum. Resistant or susceptible to Fusarium wilt?

Line 114: Move primers' sequences to a Supplemental Table.

Table 1 could be moved as Supplemental material.

Figure 1. With the legend, the figure must stand alone. Provide essential experimental details. Statistical information.

Table 2. Indicate the statistical test employed and the number of replicates (N).

Legend Figure 3. Provide some details of the experiment. What was the inoculum?

Figure 4. More details, please. What is the difference between b and c? Time of exposure to MM??

Figure 5. More details, please. What is the difference between b and c? Time of exposure to MM?? This figure should be moved to Supplemental material.

Lines 311 – 316: This is poorly explained. Provide more details for each of the enzymes tested.

Figure 6. With the legend, the figure must stand alone. A brief title, define symbols etc…Essential experimental details. Explanations of panels.

Figure 8. With the legend, the figure must stand alone. A brief title, define symbols etc…

Essential experimental details. Explanations of panels. Statistical information. Indicate the # of replicates.

Figure 9. With the legend, the figure must stand alone. Include: Essential experimental details.

Rearrange the figures:

Top Panel: Check and MM

Bottom Panel: FON and FON+MM

Figure 10. With the legend, the figure must stand alone. A brief title, define symbols etc…

Essential experimental details. Explanations of panels. Statistical information.

This figure could be moved to Supplemental Information.

Lines 391 – 394. Quite an interesting observation. Somehow, FON is suppressing defense gene expression, even in the presence of the biocontrol agent. Elaborate on this in the Discussion section.

Figure 11. With the legend, the figure must stand alone. A brief title, define symbols etc…

Figure 14. Essential experimental details. Explanations of panels. Statistical information.

With the legend, the figure must stand alone. A brief title, define symbols etc…

Essential experimental details. Explanations of panels. Statistical information.

This figure should be completed with the MOCK treatment

Discussion should be shortened as much of it repeats the results.

Discussion should center on:

Give possible mechanisms or pathways.

Compare your results with published results.

Discuss how your findings support or challenge the paradigm.

Revise carefully the manuscript for typos, grammar and syntaxis.

Reviewer 2 Report

Dear authors!

Properties of Serratia plymuthica are well known, therefore the scientific soundness of this work is rather poor. You also used very small sample size in your study, therefore the results can not be trusted

“Effect of MM on spore germination and mycelial morphology of FON”

“The bacteria-free filtrates of MM”

If filtrates were bacteria-free, you studied not the effect of MM, but the effect of filtrates. Moreover, why did you choose to study filtrates and not extracts prepared according to the method 2.10? Were the bioactive molecules you discovered in the extract also present in the filtrate?

““2.7 Growth-promoting effect of MM on watermelon seedlings””

the grout of three plants is insufficient for scientific research

“In this study, five defense-related genes (LOX, POD, PAL, C4H, and ClPR3) were se-lected to evaluate the effect of S. plymuthica MM”

Why these genes?

“The supernatant was then mixed with n-butanol, ethyl acetate, dichloromethane, and chloroform in a ratio of 1: 1 (v/v) and separated using a separation funnel.”

Why these solvents? There is data that Serratia plymuthica produce peptides with antibiotic properties. Why didn’t you extract the peptides?

“Figure 7. The observation of fluorescent cells of GFP-MM under CLSM at 5 (a), 10 (b), 15 (c) days after bacterial inoculation.”

The microscopic images have a very bad quality and the tissues are poorly prepared. Did you observe any autofluorescence? It is very common in roots due to the high content of lignin. You should also calculate average fluorescence intensity for the experimental group. How many plants did you subject to microscopy?

“Five days after inoculation, GFP-MM was observed on the surface layer of the wa-termelon root, exhibiting the highest fluorescence intensity (Figure 7a). Furthermore, GFP-tagged MM cells were observed to penetrate the interior of the watermelon root after 10 days of inoculation (Figure 7b). By day 15, the fluorescence intensity in the roots decreased and became weak. This observation indicates that S. plymuthica MM can effectively colo-nize the roots of watermelon.”

If MM effectively colonized the roots of watermelon, why fluorescence intensity became weaker?

“inhibitory rate of 74.00%)”

the bracket is redundant

“by examining the the fluorescent cells of GFP-MM”

“The” is repeated twice

Reviewer 3 Report

Li et al., in this manuscript, explored the possibility of the Serratia plymuthica MM as a biocontrol agent against Fusarium wilt of watermelon and as a plant growth promoter. 

Introduction:

Fusarium wilt of watermelon, not watermelon fusarium wilt. Disease of the crop, not crop of the disease. 

Include the percentage and numbers of yield loss to crop production. 

Results:

The figure and table titles should be stand-alone. Expand the abbreviations used in the pictures, tables, and graphs.

Add a morphological picture of MM - lines 260-261.

Figure 2 - bold/use different color to highlight strain MM in phylogenetic tree. 

The X- axis footers are missing in the graphs.

Line 290- pathogenic fungi FON

3.8 in results - since gene expression levels are relative, relative fold levels of the gene expressed should be discussed. 

The primary concern I found in the paper lines 337-339 which states the long-term stability of the MM was not quantified when this study explores the possibility of MM as a biocontrol agent and conclusions were made that it provides effective control against fusarium wilt of watermelon. Discuss/do experiments needed to further expand this. 

Line 50- quality*

Line 67 - It produces

Rewrite lines 70-71. 

Line 71 - promise*

Line 195 - defense

Punctuations were missing in few places - which made it difficult to understand what the author was trying to convey.  

Round 2

Reviewer 2 Report

I don't see any positive changes in the manuscript.

"Serratia plymuthica isolated from
different sampling environments in different regions may be different in plant
growth-promoting effect and antifungal activities"

Then you should take samples from different environments

"Because there are antifungi substances
produced by strain MM in the filtrate, and the inhibition of spore germination was
mainly caused by these antifungi substances"

Which substances, and how do you prove their presence in the filtrate?

"“Three plants were used for each
treatment, and each treatment was replicated three times” in our manuscript means
nine plants for growth-promoting effect of MM on watermelon seedlings."

nine is still a small number, which is OK for a school project only. Your experiments with seedlings are not expensive, complicated or time consuming. I can't understand why did you decide to study such a small quantity of samples.

"The fluorescence
intensity in the roots was observed 15 days after inoculation, indicating that the strain
MM can effectively colonize the roots of watermelon, although the fluorescence
intensity decreased and became weak."

I still don't see how that can be suggested an effective colonization
